# Genomic and Experimental Investigations of *Auriscalpium* and *Strobilurus* Fungi Reveal New Insights into Pinecone Decomposition

**DOI:** 10.3390/jof7080679

**Published:** 2021-08-23

**Authors:** Panmeng Wang, Jianping Xu, Gang Wu, Tiezhi Liu, Zhu L. Yang

**Affiliations:** 1CAS Key Laboratory for Plant Diversity and Biogeography of East Asia, Kunming Institute of Botany, Chinese Academy of Sciences, Kunming 650201, China; wangpanmeng@mail.kib.ac.cn (P.W.); wugang@mail.kib.ac.cn (G.W.); 2Yunnan Key Laboratory for Fungal Diversity and Green Development, Kunming Institute of Botany, Chinese Academy of Sciences, Kunming 650201, China; 3University of Chinese Academy of Sciences, Beijing 100049, China; 4Department of Biology, McMaster University, Hamilton, ON L8S 4K1, Canada; jpxu@mcmaster.ca; 5College of Life Sciences, Chifeng University, Chifeng 024000, China; tiezhiliu@aliyun.com

**Keywords:** successive decomposition, *Auriscalpium*, *Strobilurus*, carbohydrate active enzymes, competition, secondary metabolites, strobilurin

## Abstract

Saprophytic fungi (SPF) play vital roles in ecosystem dynamics and decomposition. However, because of the complexity of living systems, our understanding of how SPF interact with each other to decompose organic matter is very limited. Here we studied their roles and interactions in the decomposition of highly specialized substrates between the two genera *Auriscalpium* and *Strobilurus* fungi-colonized fallen pinecones of the same plant sequentially. We obtained the genome sequences from seven fungal species with three pairs: *A. orientale*-*S. luchuensis*, *A. vulgare*-*S. stephanocystis* and *A. microsporum*-*S. pachcystidiatus*/*S. orientalis* on cones of *Pinus yunnanensis*, *P. sylvestris* and *P. armandii*, respectively, and the organic profiles of substrate during decomposition. Our analyses revealed evidence for both competition and cooperation between the two groups of fungi during decomposition, enabling efficient utilization of substrates with complementary profiles of carbohydrate active enzymes (CAZymes). The *Auriscalpium* fungi are highly effective at utilizing the primary organic carbon, such as lignin, and hemicellulose in freshly fallen cones, facilitated the invasion and colonization by *Strobilurus* fungi. The *Strobilurus* fungi have genes coding for abundant CAZymes to utilize the remaining organic compounds and for producing an arsenal of secondary metabolites such as strobilurins that can inhibit other fungi from colonizing the pinecones.

## 1. Introduction

Decomposition of organic matter is vitally important in ecosystem processes such as carbon and nitrogen cycling, soil formation and biodiversity maintenance [1,2]. In nature, decomposition is highly dynamic with ever-changing interactions among decomposers and between substrate composition and successive development of microbial communities [3,4]. Fungi are key components of the microbial communities in most natural decompositions with different fungal species interacting with each other in multiple ways, leading to complete degradation of complex organic compounds such as lignocelluloses in wood and plant litter. Indeed, there is emerging evidence showing orderly succession among fungal members of microbial communities through the different stages of lignocellulose decomposition [5].

As one of the most important members of microbial communities in forest ecosystems, saprotrophic fungi (SPF) have diverse degradation mechanisms and play key roles in the degradations of dead organic matters [3,6]. However, due to the limited resources across space and time in most ecological niches and the presence of many (potential) competitors, fungal decomposers have evolved mechanisms to allow them successfully colonizing one to several substrates/ecological niches [7]. Those colonizing only one type of ecological niche are called ecological “specialists” while others capable of colonizing many types of ecological niches are called “generalists” [8]. There are many who are in-between the obligate specialists and broad generalists, including those that are primarily found in one ecological niche but are capable of surviving and growing in other niches [8]. Evidence for ecological specializations in fungi has been recorded since ancient times [3]. In addition, most ecological niches and substrates have successions where different fungal communities may dominate different phases of substrate decomposition [5].

There are many factors that can impact the composition and structure of saprotrophic fungal community. Among these factors, the chemical composition of substrates plays a major role [9,10]. For example, on woody substrates, the decomposition rate of polymeric lignocellulosic components changes through the decomposition process, due to changes in substrate compositions and in the types and relative abundances of different microbes and their enzymes involved in degradations [11]. Traditionally, saprotrophic fungi are broadly classified into two types, namely ligninolytic white-rot (WR) and cellulolytic brown-rot (BR), although there is a continuum between these two types [7,12]. In the process of decomposition, interaction (including competition) among fungi is likely very common, affecting the distribution, abundance, and the order of occurrence among these fungi in natural communities [13].

Different interaction strategies among species can lead to different orders of SPF colonization on substrates. Both biotic and abiotic factors can also influence their order of colonization and interactions [7,14,15]. Fungal competition on substrates is commonly classified into two major functional types: primary resource capture and secondary resource capture [7,16,17]. Success of SPF in primary resource capture mainly depends on the ability to utilize previously uncolonized resources and on their ability to resist plant derived antifungal compounds in those substrates. In contrast, success in secondary resource capture mainly relies on antagonistic mechanisms (i.e., antifungal production etc.), with different species competing with each other to obtain sufficient nutrients for survival and reproduction [7]. Often, changes in microbial communities during decomposition are related to the secretion of antagonistic enzymes and metabolites [7]. On the one hand, SPFs have the ability to secrete various carbohydrate-active enzymes (CAZymes) to decompose and utilize the major constituents such as lignin, cellulose and hemicellulose in wood and plant litter, facilitating nutrient cycling and energy flow in forest ecosystem [18,19,20]. Indeed, the compositions and characteristics of CAZymes often differ among fungi, likely shaped by characteristics of their substrates and the degree of adaptation to the specific environmental conditions [20]. Therefore, to understand decomposition, it is particularly important to study the compositions and characteristics of CAZymes to clarify the potential mechanisms for different nutritional modes, infection and substrates specificity/preference [20,21,22,23,24,25]. On the other hand, fungal secondary metabolites (SMs) are known to play crucial roles in the defense against pathogens and competitors and provide advantages for their producers and/or those who have resistant mechanisms. Along with CAZymes, fungal SMs can provide important information for understanding the chemical basis of niche specialization during decompositions [26,27].

Multiple groups of SPFs are frequently involved in plant litter decomposition. These fungi belong to diverse clades, but some of them are functionally interchangeable [26,28,29], Appendix A. In forest ecosystems, due to its extractive composition and the presence of antifungal compounds such as resin, pinecone is a specialized substrate and a unique habitat for fungi [30,31]. Indeed, only species in a few fungal genera (e.g., *Strobilurus*, *Auriscalpium*, *Baeospora* and *Mycena*) are known to colonize and decompose pinecones. Among these genera, *Auriscalpium* and *Strobilurus* are highly specialized on pinecones [30,31]. Interestingly, species of *Auriscalpium* and *Strobilurus* usually share the dead cones of the same plant species in a chronological order, with fruiting bodies of *Auriscalpium* fungi often appearing on newly fallen cones, while those of *Strobilurus* typically occurring on highly rotten cones during later stages of decomposition (Appendix A). At present, the mechanisms for their succession during pinecone decomposition are unknown.

In this study, we investigated three pinecone substrate–fungus pairs from Europe and East Asia to understand the potential mechanisms for substrate specificities and ecological succession during pinecone decomposition. The three fungal pairs as well as their substrates were *A. orientale*-*S. luchuensis* on cones of *Pinus yunnanensis*, *A. vulgare*-*S. stephanocystis* on cones of *P. sylvestris*, and *A. microsporum*-*S. pachcystidiatus*/*S. orientalis* on cones of *P. armandii*. We obtained the genome sequences of these seven fungal species and quantified the main chemical compounds during pinecone decomposition. Our analyses revealed both shared and unique features in their substrate specificity and ecological successions among these fungal pairs.

## 2. Materials and Methods

### 2.1. Material Collections, Greenhouse Planting Experiment, Fungal Strains, Media and Culture Conditions

The records on the geographical distributions and substrate information of *A. vulgare* and *S. stephanocystis* were extracted from data deposited in the Global Biodiversity Information Facility (GBIF; https://www.gbif.org/; accessed on 11 April 2019). The information on substrates and distributions of the following four species *A. orientale*, *A. microsporum*, *S. luchuensis* and *S. pachycystidiatus* was primarily from the published literature and our own observations during field investigations in China. In order to directly observe the characteristics of pinecone decomposition by *Auriscalpium* and *Strobilurus*, a planting experiment was conducted in the greenhouse using about 250 cones of *P. armandii* fully colonized by *A. microsporum* collected in Yufeng Temple (Lijiang City, Yunnan Province, China) in July 2017 where both *Auriscalpium* and *Strobilurus* existed. The reason for the selection of collection site of Yufeng Temple is that Yufeng Temple is the distribution center of fungi in *Strobilurus* and *Auriscalpium*, and the climate in this area is close to Kunming City. After we brought cones back without any specific treatment, we put them in the plastic shed and spray water regularly to ensure their humidity. During our experiment, we keep the temperature in the greenhouse consistent with the outdoor temperature of Kunming.

All seven strains used for de novo sequencing were dikaryotic strains isolated directly from the fruiting bodies from wild mushrooms. Among these, the following five species and strains were collected in Yunnan Province, China: *A. microsporum* (strain AU-H-210), *A. orientale* (strain AU-Y-A), *S. luchuensis* (strain Y-Y-2D), *S. pachycystidiatus* (strain Y-H-6C) and *S. orientalis* (strain K-1-1). The remaining two *A. vulgare* (CBS 236.39) and *S. stephanocystis* (CBS 113577) were obtained from the Westerdijk Fungal Biodiversity Institute (CBS, Fungal Biodiversity Centre in Netherlands). The strains were stored at 4 °C on MEA solid medium (1% malt extract, 0.1% peptone and 1.5% agar). Vegetative mycelia of those strains were cultivated in MEA liquid medium (1% malt extract and 0.1% peptone) in the dark at 23 °C for 7–25 days, and then the mycelia were collected for DNA extraction and whole genome sequencing [32].

### 2.2. Measurements of Pinecone Compositions

For each type of pinecone, we analyzed the complex organic compounds at two different states: (i) newly fallen cones, and (ii) cones being decomposed by fungi in the genus *Auriscalpium* and just colonized by fungi in *Strobilurus* (Appendix A). The contents of cellulose, hemicellulose and lignin were quantified using the automatic fiber analyzer according to the ANKOM 2000i instructions (https://www.ankom.com/technical-support/fiber-analyzer-a2000; accessed on 21 April 2019) in cones of *P. armandii*, *P. yunnanensis* and *P. sylvestris* at KIB (Kunming Institute of Botany, CAS) as references. The pectin contents in these samples were quantified by using the method of sulfuric acid-carbazole colorimetry [33]. Analyses for all samples were carried in decuplicate.

### 2.3. Genome Sequencing, Assembly and Protein-Coding Gene Predictions

To yield a high-quality genome assembly, the genomes were sequenced using a whole genome shotgun sequencing strategy with a combined strategy that included both the Pacific Biosciences RS II (Pacific Biosciences, Menlo Park, CA, USA) and Illumina MiSeq platforms (Illumina Inc., San Diego, CA, USA) by Biomarker Technologies Co, LTD (Beijing, China). The PacBio long reads were corrected and assembled using Canu for draft genomes [34]. FinisherSC was used to improve the contiguity of draft genomes [35], and pilon was used to polish the draft genomes with collected Illumina data by Musket [36,37]. HaploMerger2 was used to separate the two haploid sub-assemblies from the assembly [38,39]. Ab initio predictions were carried out using the reference protein domains of *Peniophora* sp. and *Armillaria ostoyae* [40,41] for *Auriscalpium* and *Strobilurus*, respectively. Based on the two high-quality sequencing datasets described above, the protein-coding gene set of genomes were refined following the GETA gene annotation method [42]. BUSCOs using database of fungi_odb9 were applied to our gene predictions.

### 2.4. Identification of Carbohydrate-Active Enzymes (CAZymes) and Lignocellulolytic Genes and Swiss-Prot Annotation

Carbohydrate-active enzymes (CAZymes) were identified by using a combination of pipelines that included the HMM and BLASTP algorithms [43]. CAZyme annotation by BLASTP algorithms used a cutoff *e-value* < 1 × 10^−5^ and coverage >20%. CAZyme annotation by HMM algorithms used a cutoff *e-value* < 1 × 10^−5^ for alignments of >80 amino acids, and for alignments of <80 amino acids, we used an *e-value* of < 1 × 10^−3^ and coverage >25%. Perl program was used to extract the annotation results that conform to the two methods as the final result. For Swiss-Prot annotation, the BLASTP algorithm was used to align the protein sequences to Swiss-Prot with *e-value* < 1 × 10^−5^ and coverage >20%. Lignocellulolytic genes (cellulase, hemicellulase, pectinase, lignin oxidase and lignin degrading auxiliary enzymes) were identified mainly by the Swiss-Prot annotation with keywords as described by Chen et al. [43] in their Appendix A. In the following analyses, lignin oxidases and lignin degrading auxiliary enzymes encoded by lignocellulolytic genes were combined into one category called ligninases [43].

### 2.5. Principal Component Analyses (PCA) and Heatmap Analyses of CAZymes and Lignocellulolytic Genes

In PCA of CAZymes, organisms are clustered with others which have the similar nutrient patterns to determine the rot patterns of fungi. Here, PCA was used to cluster WR and BR fungi based on the diversity of their CAZymes. The CAZymes number matrix of wood decay fungi (WDF) from the following species was used as input: *Galerina marginata* (abbreviated as “gama”), *A. orientale* (“auor”), *Heterobasidion annosum* (“hean”), *Stereum hirsutum* (“sthi”), *A. vulgare* (“auvu”), *Punctularia strigosozonata* (“pust”), *A. microsporum* (“aumi”), *Fomitiporia mediterranea* (“fome”), *S. pachycystidiatus* (“stpa”), *S. luchuensis* (“stlu”), *Dichomitus squalens* (“disq”), *Trametes versicolor* (“trve”), *Phanerochaete chrysosporium* (“phch”), *S. stephanocystis* (“stst”), *Phanerochaete carnosa* (“phca”), *S. orientalis* (“stor”), *Wolfiporia cocos* (“woco”), *Gloeophyllum trabeum* (“gltr”), *Fomitopsis pinicola* (“fopi”), *Dacryopinax primogenitus* (“dapr”), *Serpula lacrymans* (“sela”) and *Coniophora puteana* (“copu”) [12,44,45,46,47,48,49].

Among these species, phca, phch, hean, sthi, gama, disq, trve, fome and pust belonged to WR fungi, while copu, gltr, fopi, woco, dapr and sela belonged to BR fungi [12,43]. The online OmicShare tools (http://www.omicshare.com/tools; accessed on 14 August 2019) were used to generate the illustrations, heatmaps and other outputs from PCA and statistical analysis. Adobe Photoshop (PS) and Adobe Illustrator (AI) were used for image editing and finalization.

### 2.6. Analyses of Enzymes on Resin Decomposition

By adding latex to media, Oghenekaro et al. (2020) identified that some genes had elevated expressions in *R. microporus*. Latex is a complex emulsion that includes a diversity of chemicals such as proteins, alkaloids, carbohydrates, oils, tannins, and resins that coagulate on exposure to air [50,51]. It is commonly produced by plants after tissue injury and serve as a defense against pathogens and pests [50,51]. Here we used the high expressed genes induced by latex treatment in *R. microporus* to infer the distributions of these genes in the seven fungi analyzed in our study. Using those genes in *R. microporus* as references, we identified the homologous genes in our genomes through BLASTP algorithms with a cut off *e-value* < 1 × 10^−5^ and coverage >20%.

### 2.7. Secondary Metabolites (SMs) in Auriscalpium and Strobilurus

SMs genes are frequently located in gene clusters of microorganisms [52], and they may have important physiological and ecological significances with antifungal, antibacterial, antitumor, antiviral, antialgal, immune-suppressive and other biological effects [53]. Here, for our seven genomes, SM gene clusters were determined using a web-based analysis platform named AntiSMASH fungal 6.0 (https://fungismash.secondarymetabolites.org/#!/start; accessed on 13 August 2019) [54].

### 2.8. Species Tree Construction, Divergence Time Estimation and Gene Family Expansions of CAZymes Analysis

Orthologous genes and single-copy orthologous genes among 22 publicly available fungal genomes above were identified by using software of OrthoMCL [55,56] with same parameters of Chen et al. [43]. The generated proteomes and corresponding coding sequences were used as input to phylogenomic and comparative genomic analyses. We constructed a phylogenomic tree included genomes with predicted proteome clusters generated from the comparative analysis of 22 available fungal genomes, with *Dacryopinax primogenitus* as the outgroup. Phylogenomics analysis was conducted based on the framework proposed by Chen et al. [43] using software of ProtTest [57], MAFFT v7.158b [58], Gblocks 0.91b and RAxML-8.0.26 [59]. For divergence time estimation with r8s v1.80 [60] with same parameters of Chen et al. [43], one fossil calibration points and three secondary calibration point [45] were fixed in the molecule clock analysis: the most recent common ancestor (MRCA) of *Serpula lacrymans* and *Coniophora puteana* were diverged at 104.23 million years ago (MYA); the MRCA of *Punctularia strigosozonata* and *Gloeophyllum trabeum* were diverged at 170.42 MYA; the MRCA of *Stereum hirsutum* and *Heterobasidion annosum* were diverged at 100.94 MYA and the MRCA of *Fomitiporia mediterranea* and *Dacryopinax primogenitus* were diverged at 349.72 MYA. The orthologous gene family expansions and contractions were calculated by CAFE with p-values less than 0.05 based on the ultrametric tree [61]. When we got the expanded gene families, we selected the expanded gene families belonging to CAZymes for subsequent analysis.

## 3. Results

### 3.1. Ecological Characteristics of Fungi in Auriscalpium and Strobilurus

The temporal and trophic niches as reported in the literature and our own observations for *Auriscalpium* and *Strobilurus* fungi in this study are summarized in Figure 1, Appendix A and Appendix A. The data showed clear differences between these two groups of fungi in their temporal distributions during pinecone decomposition (Figure 1, Appendix A; Appendix A). Specifically, fruiting bodies of *A. vulgare* are mostly found on newly fallen cones of *P. sylvestris*, *P. pinaster*, *P. halepensis* and *P. mugo* in September and October, while those of *S. stephanocystis* mostly appear on highly rotten cones already decomposed by *A. vulgare* in May. Similarly, *A. orientale* mainly appears on newly fallen cones of *P. yunnanensis*, *P. tabuliformis*, *P. densiflora*, *P. densata*, *P. massoniana* and *P. hwangshanensis* in August, while *S. luchuensis* on highly rotten cones already decomposed by *A. orientale*. In addition, *A. microsporum* mostly produces fruiting bodies in September on newly fallen cones of *P. armandii* and with fruiting bodies of *S. pachycystidiatus* found in June on cones newly decomposed by *A. microsporum*, and *S. orientalis* in October on highly decomposed cones by *A. microsporum* and/or *S. pachycystidiatus*. Although the time of fruiting was different, the preferences of the two genera for the substrates with different degrees of decay were not affected.

The ecological characteristics of colonization and fruiting by *Auriscalpium* and *Strobilurus* fungi on pinecones as observed in the field were similarly found in our greenhouse fruiting experiment (Appendix A). There are hyphae or spores of *S. pachcystidiatus* and *S. orientalis* in the cones occupied by *A. microsporum* collected from the field. Under the greenhouse culture condition, fruiting bodies of *A. microsporum* formed first from July to September, then *S. pachcystidiatus* appeared from May to July over the next 3 years, and finally *S. orientalis* appeared from October to December. Together, these greenhouse observations are consistent with our field observations on the ecological successions of these fungi on pinecones (Appendix A). However, because the environmental humidity in the greenhouse is stable, the fruiting time of fungi in *Auriscalpium* and *Strobilurus* is relatively continuous.

### 3.2. Genome Sequencing, Data Preprocessing, Assembly, General Genome Features, Protein-Coding Gene Prediction and Functional Annotation

The general features of the seven genomes that we sequenced are summarized in Table 1. Raw data were generated with Pacbio sequencing and Illumina sequencing with coverage 85.84–386.08× and 101.04–206.85×, respectively (Appendix A). Among the *Auriscalpium* species, *A. vulgare* had the largest genome (51.68 Mb), followed by *A. orientale* (45.40 Mb), and *A. microsporum* had the smallest genome (43.46 Mb) (Table 1). Among the species of *Strobilurus*, *S. pachycystidiatus* had the largest genome (51.82 Mb), followed by *S. orientalis* (51.25 Mb) and *S. luchuensis* (46.71 Mb), and *S. stephanocystis* had the smallest genome (42.38 Mb) (Table 1). The sets of annotated protein-coding genes in the seven assembled genomes were estimated to be 92.4–100% complete (Appendix A). Among the *Auriscalpium* species, *A. orientale* had the most coding genes (16,958), followed by *A. microsporum* (15,333), and *A. vulgare* had the least coding genes (13,636). Among the taxa of *Strobilurus*, *S. orientalis* has the most coding genes (18,509), followed by *S. pachycystidiatus* (18,157) and *S. luchuensis* (16,796), and *S. stephanocystis* had the least number of coding genes (16,439).

### 3.3. Identification of CAZymes and Lignocellulolytic Genes in Fungi of Auriscalpium and Strobilurus

The number of CAZymes in *A. vulgare*, *A. microsporum*, *A. orientale*, *S. stephanocystis*, *S. luchuensis*, *S. pachycystidiatus* and *S. orientalis* were 450, 425, 464, 532, 542, 583 and 621, respectively (Figure 2b; Appendix A). Our statistical analyses revealed that the average number of CAZymes in *Auriscalpium* fungi is significantly lower than that in *Strobilurus* such as AA5, CBM18, CBM67, CE12, CE16, CE4, CE8, GH105, GH12, GH127, GH128, GH13, GH135, GH16, GH17, GH18, GH27, GH28, GH29, GH35, GH43, GH45, GH5, GH53, GH55, GH71, GH76, GH93, GT1, GT15, GT17, GT33, GT4, GT8, PL1 and PL4 (Appendix A). The average number of CAZymes in *Auriscalpium* fungi is significantly more than that in *Strobilurus* only in seven gene families including AA2, GH2, GH3, GH15, GH31, GH109 and PL8 (Appendix A). Similarly, in the statistical analyses, the comparison of CAZymes between *Auriscalpium* and other WR fungi, and the comparison of CAZymes between *Strobilurus* and other WR fungi are shown in Appendix A, respectively. The average number of CAZymes in *Auriscalpium* fungi is significantly more than that in other WR fungi only in six gene families, however the average number of CAZymes in Strobilurus fungi is significantly more than that other WR fungi in 31 gene families (Appendix A).

The number of predicted lignocellulolytic genes in *A. vulgare*, *A. microsporum*, *A. orientale*, *S. stephanocystis*, *S. luchuensis*, *S. pachycystidiatus* and *S. orientalis* were 112, 97, 111, 107, 106, 122 and 111, respectively (Appendix A). Our analyses demonstrated that the number of genes coding for AA2, GH3 and GH7 in *Auriscalpium* are significantly higher than those in *Strobilurus*, while the number of genes coding for AA1, CE8, GH12, GH5_5, GH45 and PL1_4 in *Strobilurus* are significantly higher than those in *Auriscalpium* (Appendix A). For the fungi growing on cones of the same pine species, the proportions of genes encoding ligninases and hemicellulases were higher in *Auriscalpium* species than those in their corresponding *Strobilurus* species. In contrast, the proportions of genes encoding cellulases and pectinases were lower in *Auriscalpium* than those in the *Strobilurus*. For fungi living on cones of *P. armandii*, *A. microsporum* and *S. pachycystidiatus* living on relatively newly fallen cones also showed a similar pattern but the difference was less obvious than those in other two fungal pairs. However, *S. orientalis* with a preference of living on more rotten cones showed more obvious differences in their respective enzymes compared with *A. microsporum*.

### 3.4. PCA and Heatmap Analyses of CAZymes and Lignocellulolytic Genes

Results of the PCA of CAZymes and lignocellulolytic genes showed a clear separation of WR and BR fungi along PCA1 (Figure 2a and Appendix A). All the fungi in *Auriscalpium* and *Strobilurus* cluster together with other WR fungi, consistent with their potent lignin-decomposing ability (Figure 2a and Appendix A). The heatmap analyses showed that there are unique enzymes within each of the 22 fungi analyzed, which likely reflect the diversity of their preferred substrates (Figure 2c and Appendix A). In the separate analyses of the two genera, it is apparent that fungi in the two genera showed complementary profiles of carbohydrate-active enzymes (CAZymes) (Figure 3a and Figure 4g,h). The results clearly illustrate that there are remarkable differences in the overall pattern of genes between two different genera. At the same time, it demonstrates that the type and number of these CAZymes in *Auriscalpium* and *Strobilurus* are different among different fungi, which may partly explain the cause of the substrate specificity (Appendix A). The results of the lignocellulolytic genes are overall consistent with the results of CAZymes (Appendix A). These lignocellulolytic genes are also specific among genera, and there are great differences in the type and number of enzymes among different genera, and each has a set of different enzyme system (Figure 4g,h and Appendix A).

### 3.5. Composition Analyses of Pinecones

Among newly fallen cones, that of *P. yunnanensis* showed the highest contents of lignin and hemicellulose, but the least amounts of cellulose and pectin. In contrast, the cones of *P. armandii* showed the highest contents of cellulose and pectin, but the least amount of lignin (Figure 3b; Appendix A). In addition to the least amount of hemicellulose, the other components in the cone of *P. sylvestris* are between those in the cones of the other two species (Figure 3b; Appendix A). Compared with the newly fallen cones, the cones already decomposed by corresponding *Auriscalpium* fungi had reduced relative proportions of lignin and hemicellulose and increased relative proportions of cellulose and pectin (Figure 4a–c; Appendix A).

### 3.6. Enzymes Related to Resin Decompositions

Among the top 10 enzymes showing elevated expressions in the presence of latex in *R. microporus*, our analyses revealed that four of them had an overall greater numbers of genes coding for these enzymes in *Auriscalpium* fungi than in *Strobilurus* in the top No. 1, 2, 6 and 10 in *Auriscalpium* is more than those in *Strobilurus* (Table 2). Overall, among all the genes, the number of genes coding for peptidase S8 (No. 1) and peptidase S53 (No. 6) in *Auriscalpium* were significantly higher than that in *Strobilurus* (*p* < 0.05), while the other genes were not significantly different. Of the genes coding for peptidase S8, the average number in *Auriscalpium* is 20.33, while only 5.4 in *Strobilurus* (Table 2). In genes coding for peptidase S53, the average numbers in *Auriscalpium* and *Strobilurus* were 3.7 and 1.2, respectively (Table 2).

### 3.7. SM Clusters in Strobilurus and Auriscalpium

Terpenes, NRPS (nonribosomal peptide synthetase)-like compounds, and siderophores are among the main groups of SMs shared by fungi in *Auriscalpium* and *Strobilurus* and the putative genes coding for enzymes involved in their syntheses are summarized in Table 3. Our analyses showed that T1PKS (type I polyketide synthases)-NRPS like and betalactone existed solely in *Auriscalpium*, while T1PKS-terpene, strobilurin, T1PKS, and indole alkaloids were only found in *Strobilurus* (Table 3). Our results revealed that the average number of gene clusters for SMs in *Auriscalpium* fungi was fewer (average 18.33) than that in *Strobilurus* (average 21.25). As expected, each species of *Strobilurus* harbors one SM clusters of strobilurin (Table 3). Fungi in *Auriscalpium* contain very few gene clusters for NRPS-like but abundant for terpenes, while fungi in *Strobilurus* are the opposite (Table 3). In summary, the number of genes predicted for the synthesis of NRPS-like, T1PKS, strobilurin and indole alkaloids in *Strobilurus* fungi is higher than those in *Auriscalpium* taxa.

### 3.8. Result of Species Tree Construction, Divergence Time Estimation and Gene Family Expansions of CAZymes Analysis

Maximum-likelihood phylogenetic analysis using concatenated nucleotide sequences of the 1519 single-copy orthologs identified through OrthoMCL analysis present in all species of 22 fungi. Most of the nodes have high support rate, which proves that our phylogenetic tree can well analyze the phylogenetic and evolutionary relationship between species. The result of divergence time estimation is shown in Figure 5a.

Among those expanded Ortholog Cluster Groups (OCGs) examined using CAFE since their most recent common ancestor (MRCA) bifurcated, *S. stephanocystis* contained 3 OCGs of CAZymes including OCG97 (AA3_3) (gene number (GM) 5), OCG128 (AA7) (GM 4) and OGC281 (GH18) (GM 4), *S. luchuensis* contained 6 OCGs of CAZymes including CG1483 (AA2) (GM 3), OCG128 (AA7) (GM 2), OCG338 (AA7) (GM 3), OCG509 (AA7) (GM 2), OCG847 (GH11) (GM 4) and OCG73 (GH35) (GM 8), *S. pachycystidiatus* contained 6 OCGs of CAZymes including OCG24 (AA1_1) (GM 9), OCG128 (AA7) (GM 4), OCG6763 (AA7) (GM 3), OCG 119 (GH7) (GM 8) and OCG489 (GH18) (GM 4), *S. orientalis* contained 6 OCGs of CAZymes including OCG82 (AA3_2) (GM 6), OCG338 (AA7) (GM 6), OCG665 (AA7) (GM 5), OCG281 (GH18) (GM 6), OCG436 (GH28) (GM 3) and OCG73 (GH35) (GM 10), *A. microsporum* contained 5 OCGs of CAZymes including OCG14 (AA3_2) (GM 13), OCG97 (AA3_3) (GM 6), OCG338 (AA7) (GM 4), OCG119 (GH7) (GM 6), and OCG847 (GH11) (GM 5), *A. vulgare* contained 5 OCGs of CAZymes including OCG14 (AA3_2) (GM 19), OCG624 (AA3_2) (GM 6), OCG128 (AA7) (GM 12), OCG509 (AA7) (GM 6), and OCG910 (AA9) (GM 3), and *A. orientale* contained 3 OCGs of CAZymes including OCG24 (AA1_1) (GM 13), OCG130 (AA4) (GM 12), and OCG259 (GH79) (GM 3) (Figure 5b; Appendix A). Those expanded OCGs of other species shown in Figure 5b and Appendix A.

## 4. Discussion

### 4.1. Successive Decomposition of Pinecones by Fungi of Auriscalpium and Strobilurus

The successive decomposition of substrates by microbial communities is a common phenomenon [28,62]. Often, the microbial community structure, including the relative abundances of saprotrophic fungi, changes significantly during the successive decomposition process [63]. Though occupying the same ecological niche, species in these communities may develop unique but complementary strategies to partition the resources in the substrates, leading to temporal niche differentiation and divergence [64]. Here, part of the resource partition is temporal changes of fungi with different fungi use different sets of nutrients within the pinecone. Similar phenomena have been found in other substrates such as plant litters and deadwoods [3,13,65]. In the process of biodegradation, microbial coordination with different ecological strategies and certain orders are evident [65,66]. For example, successions of fungi in temperate forests were considered to be reflected in sugar utilizing fungi, followed by wood structural decaying fungi and finally residual decaying fungi in some cases [67]. This change may be explained in part by nutrients released by the primary decomposers that enabled the colonization of secondary decomposers [7]. However, the successive decomposition of substrates, such as deadwood and plant litter, requires the action and interaction of many fungi with their fungal community showing a high degree of complexity [3,68]. For example, Zhang and Wei [29] had carried out relevant research on fungi in the same forest, in which different fungi will appear on rotten wood in the same state, or even on the same rotten wood. At the same time, the same kind of fungi can also exist in different periods of rotten wood (Appendix A). Some fungi can only appear in one period, but most fungi can produce fruiting bodies at several stages of rotten wood [29], Appendix A. Similarly, Niemela et al. [28] reported the succession of more than one hundred species of lignicolous Basidiomycetes on fallen trunks in *Picea obovata* and *P. sylvestris*. Our study revealed that fungi in *Auriscalpium* and *Strobilurus* possess clear differences in the type and number of CAZymes and lignocellulolytic genes (Figure 2c, Figure 3a, Figure 4g,h and Appendix A). Our results indicate that even though they colonize the same pinecones, there are significant divergence and niche differentiation in the utilization of substrates in pinecones between the fungi of the two genera, which leads to the dynamic changes of their colonization on the pinecones.

During the initial decomposition of pinecones, *Auriscalpium* fungi are the primary colonizers, likely related to their ability to break down resin and their strong capacity to decompose lignin and hemicellulose (Figure 4a–c; Table 2). Such abilities are common among WR fungi [69]. For example, the fungi of *Ceriporiopsis subvermispora*, *Phellinus pini*, *Ganoderma australe*, and *Phlebia tremellosa* specifically degrade lignin and hemicellulose among WR fungi [70]. The most compelling evidence supporting the early colonizing ability of *Auriscalpium* fungi is that their number of peptidases S8 and S53 is far greater than that in *Strobilurus* fungi. Peptidases S8 and S53 are among the top 10 most up-regulated enzymes in *R. microporus* in the presence of latex [50], Table 2. The genomic evidence is consistent with *Auriscalpium* fungi capable of colonizing newly fallen cones and decomposing proteins in resin rapidly. Polo et al. [71] showed that lignin and hemicellulose are in the outermost layer of plant cell wall which prevents the cellulolytic enzymes reaching the cellulose and protect plants from microbes. Our analyses demonstrated that the number of genes coding for lignin oxidases (AA2) and hemicellulase (GH3) in *Auriscalpium* are significantly higher than those in *Strobilurus* (Figure 4g,h; Appendix A), and these genes may be related to lignin and hemicellulase decompositions in the outermost layer of pinecones. Similarly, in the analysis of gene family expansions, we found that the number of lignin related decomposition gene families was significantly higher in *Auriscalpium* than those of fungi in *Strobilurus*. For example, there are six gene families related to lignin decomposition expanded in *Auriscalpium*: *A. microsporum* including OCG14 (AA3_2) (GM 13) and OCG97 (AA3_3) (GM 6), *A. vulgare* including OCG14 (AA3_2) (GM 19) and OCG624 (AA3_2) (GM 6), and *A. orientale* including OCG24 (AA1_1) (GM 13) and OCG130 (AA4) (GM 12), while only four gene families related to lignin decomposition expanded in *Strobilurus*: *S. stephanocystis* including OCG97 (AA3_3) (GM 5), *S. luchuensis* including CG1483 (AA2) (GM 3), *S. pachycystidiatus* including OCG24 (AA1_1) (GM 9), and *S. orientalis* including OCG82 (AA3_2) (GM 6) (Figure 5b; Appendix A). Once the outer layer is breached, the condition is now more favorable for the subsequent invasion of *Strobilurus* fungi. With increasing decay, the nutritional composition, physical structure, chemical composition and other aspects of the pinecones have changed, which result in the succession changes of fungal community.

After decomposition by *Auriscalpium* fungi, the proportions of lignin and hemicellulose in the pinecone would decrease and those of cellulose and pectin proportionally would increase (Figure 4a–c). The subsequent colonization by *Strobilurus* fungi relies on the residual components of the cones suitable for their growth and replacing the corresponding fungi of *Auriscalpium* (Figure 6). Similarly, comparing *Auriscalpium* and *Strobilurus* grown on the same pinecone, fungi of *Strobilurus* show decreasing trends of in the number of genes coding for ligninases and hemicellulases, but with higher number of genes coding for cellulase and pectinase, which is broadly consistent with the changes of cone components (Figure 4a–f). *Strobilurus pachcystidiatus* and *A. microsporum* also show the same pattern, but the differences are not particularly evident, which may relate to the fact that both could grow on newly fallen cones. However, *S. orientalis* grew on the highly rotten cones after decomposition by *A. microsporum* or *S. pachcystidiatus* and it showed a more obvious decrease in the number of ligninase-encoding genes and an increase in the number of cellulase-encoding genes than *S. pachcystidiatus* (Figure 4f). Accordingly, in the analysis of gene family expansions, we found that the number of GH gene families of fungi in *Strobilurus* was significantly higher than those of fungi in *Auriscalpium*. For example, there are eight gene families related to GH expanded in *Strobilurus*: *S. stephanocystis* including OGC281 (GH18) (GM 4), *S. luchuensis* including OCG847 (GH11) (GM 4) and OCG73 (GH35) (GM 8), *S. pachycystidiatus* including OCG 119 (GH7) (GM 8) and OCG489 (GH18) (GM 4), and *S. orientalis* including OCG281 (GH18) (GM 6), OCG436 (GH28) (GM 3) and OCG73 (GH35) (GM 10), while only three gene families related to GH expanded in *Auriscalpium*: *A. microsporum* including OCG119 (GH7) (GM 6), and OCG847 (GH11) (GM 5) and *A. orientale* including OCG259 (GH79) (GM 3) (Figure 5; Appendix A). These gene families of GHs are related to the decomposition of cellulose, hemicellulose and pectin, respectively. For example, GH7 is related to cellulose decomposition, GH11 and GH35 are related to hemicellulose decomposition, and GH28 is related to pectin decomposition. All the above results show that the fungi in *Strobilurus* may have a good decomposition effect on utilizing the remaining organic compounds of the decomposed cones.

In addition, in the field, we observed that the fungi of *Auriscalpium* can decompose cones independently, especially in tropical areas, but the successive decomposition of the two genera is more common. However, we did not observe the decomposition of cones by fungi of *Strobilurus* independently. In each distribution areas of fungi in *Strobilurus*, fungi in *Auriscalpium* could be collected in different periods, and the fungi in *Strobilurus* collected all grow on the cones with high degree of decay. For successive decomposition of *P. armandii*’s cones, in addition to the most common combination of *A. microsporum*-*S. pachcystidiatus*-*S. orientalis*, we also observed the combinations of *A. microsporum*-*S. pachcystidiatus* and *A. microsporum*-*S. orientalis*. Therefore, various situations may occur in the field (Figure 6). Although *Strobilurus* fungi always appear on cones with a high degree of decomposition, however, the results of our field observation show that compared with *Auriscalpium* fungi, the *Strobilurus* fungi can occupy the cone for a long time and fully decompose the cone. There have been reports on the positive correlation between the large amount of CAZymes in the genome and the degradation of plant biomass [72], so we speculate that fungi in *Strobilurus* are the main decomposer with the type and number of CAZymes in *Strobilurus* being richer than those of *Auriscalpium* (Figure 2b,c and Figure 3a). In the CAZymes comparisons between the two genera, only seven CAZyme gene families have significantly more genes in the *Auriscalpium* fungi than in *Strobilurus*, while the other 36 gene families have more genes in *Strobilurus* fungi than in *Auriscalpium* fungi (Appendix A), which broadly supports that the subsequent decomposers are main components of substrate decompositions [17].

### 4.2. Fungal Competition of Auriscalpium and Strobilurus on Pinecones

Successive decomposition may be affected by fungal competition. Fungal competitions of *Auriscalpium* and *Strobilurus* on substrates correspond to the two functional types: primary resource capture and secondary resource capture, respectively [7], so that the two genera appear orderly on the cones. On newly fallen cones, there may be hundreds of fungi competing to colonize cones on the surface, which is similar to the situation reported in woods [73]. However, the most obvious obstacles for fungi on newly fallen cones are the lack of easily assimilated nutrient matrixes such as lignin and hemicellulose, and the presences of inhibitory substances such as resin, etc. [74]. Due to their ability to breakdown these resistant substances, *Auriscalpium* can gain a competitive edge with other microorganisms in microbial community on newly fallen cones.

With the increase of the decomposition levels, the competitive pressures on the cones gradually increase attributed to the disappearance or reduction of antibacterial or antifungal substances. On one hand, the cones decomposed by the *Auriscalpium* provide suitable conditions for *Strobilurus* growth. During this period, the fruiting bodies of *Strobilurus* and *Auriscalpium* can co-exist on the same cone (Appendix A). On the other hand, fungi of *Strobilurus*, as aggressive competitors, produce toxins (e.g., strobilurin) to enhance its competitiveness with *Auriscalpium* and other microorganisms (Appendix A). For nutrients and spaces, the antagonists can either interact directly with the competitors [75], or secrete antibiotics to suppress competitors [76]. Some studies on antifungal SMs and enzymes produced by fungi with antagonists have been conducted extensively in vitro [77,78]. Through SM clusters analyses, we find that the number of SM clusters of the genus *Auriscalpium* is fewer than that of *Strobilurus* (Table 3), while all fungi of *Strobilurus* contain a fungicidal strobilurin cluster whose derivatives have broad spectrum antifungal activity, and, thus, are widely used as biological fungicide [79]. In addition, the number of NRPS-like in the genomes of *Strobilurus* is higher than those in *Auriscalpium* (Table 3). Most of the NRPS, PKS and their combinations have antibacterial and antifungal activities [80], which indicates that the fungi in *Strobilurus* have stronger antibacterial and antifungal activities than those in *Auriscalpium*. Aside from the SMs, other factors such as microclimate, the size and quality of nutrient sources can shift the balance between fungal decomposer groups [7]. Thus, in the natural environment, the outcomes of interactions between fungi are variable, leading to differences in community structures.

## 5. Conclusions

Our results reveal that there are differentiations of temporal and trophic niches for fungi of *Auriscalpium* and *Strobilurus*. The decompositions of pinecones are frequently completed by these two groups of fungi through successive colonization, occupying the same physical niche but at different times. The primary colonizers were the fungi of *Aurscalpium* and the secondary ones were *Strobilurus* (Figure 6). The CAZymes of the two groups of fungi are highly unique but also complementary, leading to the complete biodegradation pinecones. For future research directions, we will clearly delineate successional timelines between *Auriscalpium* and *Strobilurus* to inform the mechanisms of successional decomposition through controlling expression profiles based on these timelines. Further associating these profiles with realized relative loss in the cones over the time course has the potential to link critical successional dynamics to functional outcomes.

## Figures and Tables

**Figure 1 jof-07-00679-f001:**
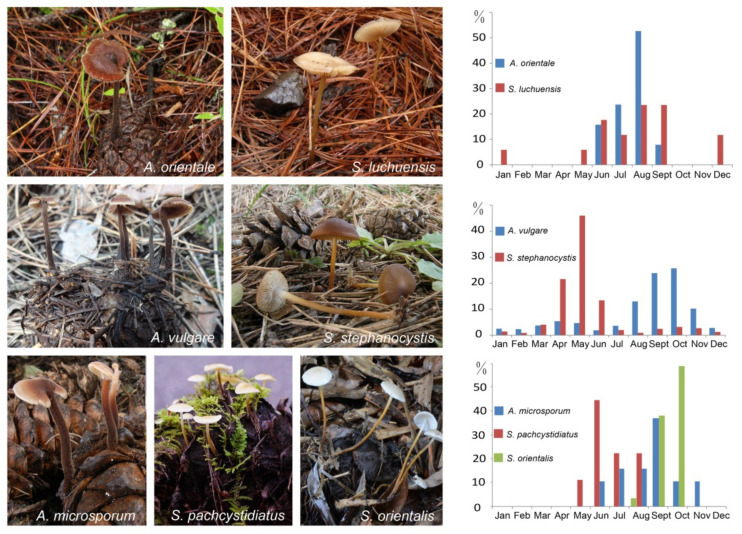
Basidiomata and percentage of monthly occurrences of fruiting bodies of *Auriscalpium* and *Strobilurus* on pinecones. The basidiomata of *Auriscalpium* usually on newly fallen cones on the ground. The basidiomata of *Strobilurus* usually on highly rotten cones under the ground. For the graphs, the *x*-axis shows months and the *y*-axis shows percentage of monthly occurrences of collected or observed fruiting bodies on pinecones in different areas and years in Appendix A.

**Figure 2 jof-07-00679-f002:**
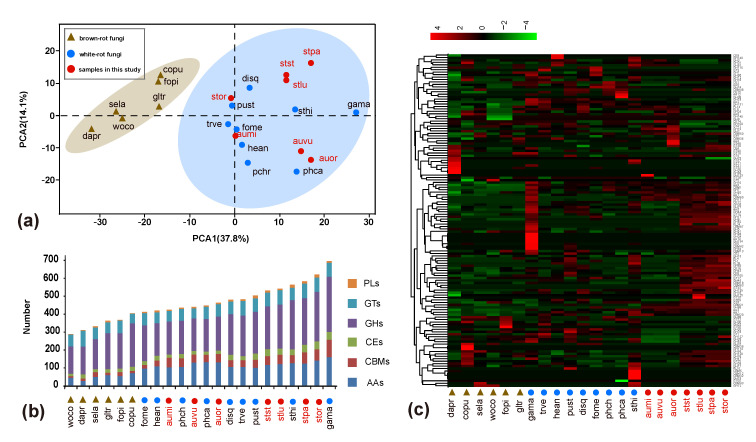
The distributions of genes encoding carbohydrate-active enzymes (CAZymes) in *Auriscalpium* and *Strobilurus* fungi. (**a**) wood decay fungi (WDF) plotted on the first two principal components from principal component analyses (PCA) of CAZymes. (**b**) Comparative analyses of CAZymes associated with lignocellulose decomposition. AAs: Auxiliary Activities; CBMs: Carbohydrate-Binding Modules; CEs: Carbohydrate Esterases; GHs: Glycoside Hydrolases; GTs: Glycosyl Transferases; PLs: Polysaccharide Lyases. (**c**) Heatmap analysis of CAZymes showing the distributions of CAZymes among different fungi. Numbers of family members in each genome are demonstrated. Overrepresented (+4 to 0) and underrepresented (0 to −4) numbers are depicted as scores for each line in heatmap. The clustering on the left involves gene families with the same pattern in number. On the right is the name of the gene family.

**Figure 3 jof-07-00679-f003:**
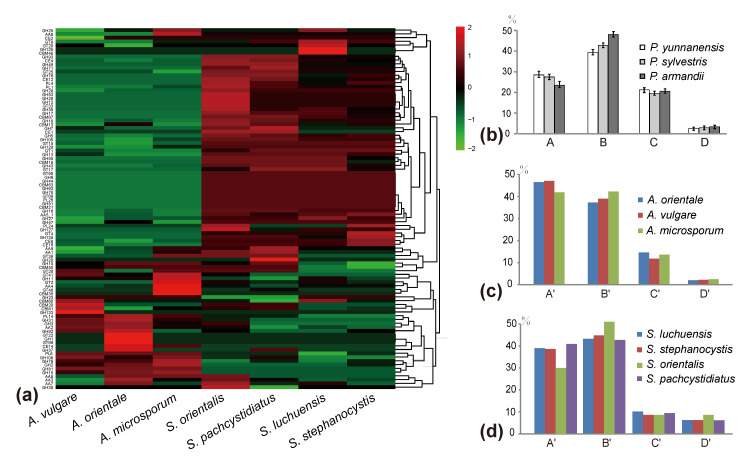
Comparisons of the carbohydrate-active enzymes (CAZymes) in *Auriscalpium* and *Strobilurus*, the proportions of four major chemical components in different cones, and the proportions of different enzymes encoded by lignocellulolytic genes in the two fungi groups. (**a**) Heatmap analysis of seven fungi in *Auriscalpium* and *Strobilurus* of CAZymes. (**b**) Four components of three different newly fallen cones. A, B, C and D in *x*-axis represent lignin, cellulose, hemicellulose and pectin, respectively. The *y*-axis represents the proportion of the four components. (**c**) The proportions of lignocellulolytic genes in different fungi of *Auriscalpium*. A’, B’, C’, D’ in *x*-axis represent ligninase, cellulose, hemicellulose, pectinase, respectively. The *y*-axis represents the proportion of the four types of enzymes; the same below. (**d**) The proportions of lignocellulolytic genes in different fungi of *Strobilurus*.

**Figure 4 jof-07-00679-f004:**
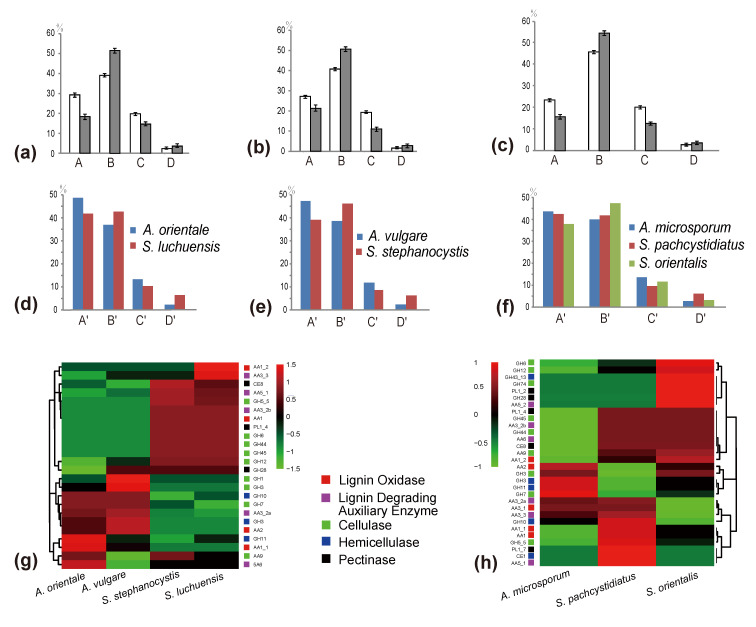
The changes of four major chemical components of cones before and after decomposition, and the relationship to differences of the numbers of lignocellulolytic genes between *Strobilurus* and *Auriscalpium* fungi colonizing the same pinecones. (**a**) The proportion of four components in cones of *P. yunnanensis* before and after the decomposition by *A. orientale*. A, B, C and D in *x*-axis represent lignin, cellulose, hemicellulose and pectin, respectively. The *y*-axis represents the proportion of the four components. White represents undecomposed cones, while gray represents decomposed cones. the same below. (**b**) The proportion of four components in cones of *P. sylvestris* before and after the decomposition by *A. vulgare*. (**c**) The content changes of four components in cones of *P. armandii* before and after being decomposed by *A. microsporum*. (**d**) Comparison of lignocellulolytic genes. A’, B’, C’ and D’ in *x*-axis represent ligninase, cellulose, hemicellulose and pectinase, respectively. The *y*-axis represents the proportion of the four types of enzymes in *A. orientalis* and *S. luchuensis* grown on cones of *P. yunnanensis*. The same below. (**e**) Comparison of lignocellulolytic genes in *A. vulgare* and *S. stephanocystis* on cones of *P. sylvestris*. (**f**) Comparisons of lignocellulolytic genes of *A. microsporum*, *S. pachcystidiatus* and *S. orientalis* grown on cones of P. armandii. (**g**) Heatmap analysis of lignocellulolytic genes of four fungi on cones of P. subgenus *Pinus*. (**h**) Heatmap analysis of lignocellulolytic genes of three fungi on cones of P. subgenus *Strobus*.

**Figure 5 jof-07-00679-f005:**
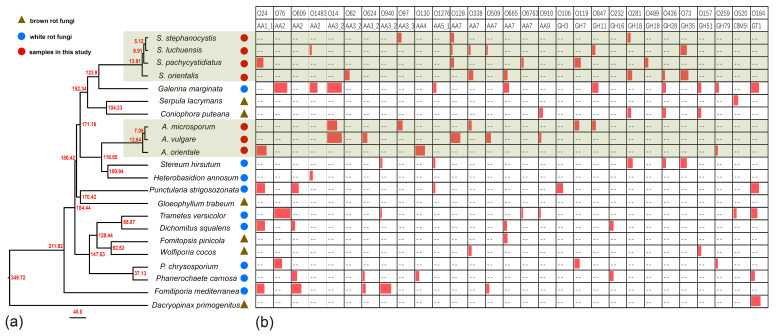
Phylogenetic tree of fungi in *Auriscalpium* and *Strobilurus* with other 15 fungal species and gene family expansions of CAZymes using CAFE. (**a**) The topology of the phylogenetic tree was constructed by the maximum likelihood method and divergence time estimation using r8s. Their divergence time was marked on the phylogenetic tree with time scale being shown by MYA (million years ago). (**b**) gene family expansions of CAZymes in 22 fungi with p-values less than 0.05. “O” in the first row of the table represents OCG (Ortholog Cluster Group); The second row of the table represents the gene family information of CAZymes. The length of the red rectangle represents the quantity; “--“ represents that the gene family has not been significantly expanded in this species.

**Figure 6 jof-07-00679-f006:**
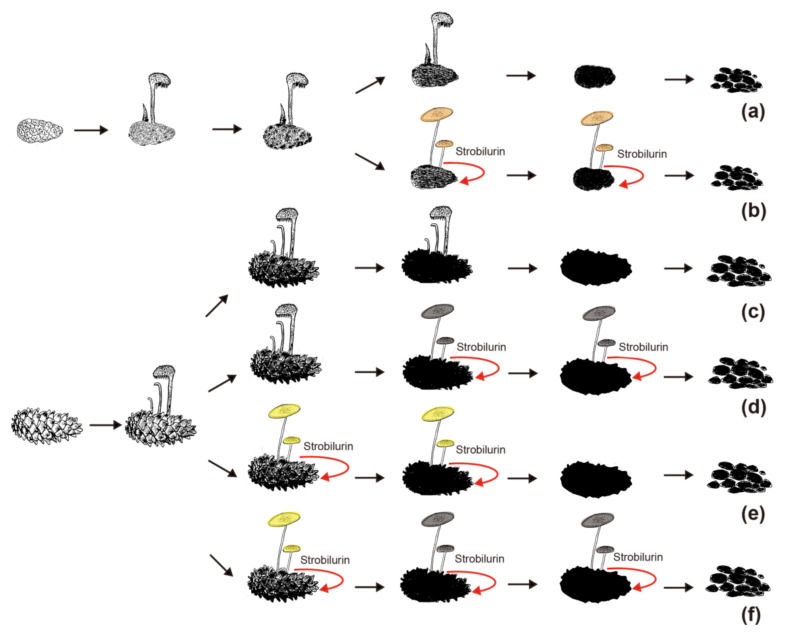
Types of successive decomposition of pinecones observed in the field. (**a**,**b**) The decomposition of cones of *P*. subgenus *Pinus*. (**a**) The cones decomposed by *A. vulgare* or *A. orientale* independently. (**b**) Successive decomposition and competition of *A. orientale*/*S. luchuensis* and *A. vulgare*/*S. stephanocystis* on cones of *P*. subgenus *Pinus*. The yellow brown pileus represents *S. luchuensis* or *S. stephanocystis*. (**c**–**f)** The decomposition of cones of P. subgenus *Strobus*. (**c**) The cones decomposed by *A. microsporum* independently. (**d**) Successive decomposition and competition of *A. microsporum* and *S. orientalis* on cones of *P*. subgenus *Strobus*. The gray brown pileus represents S. orientalis. (**e**) Successive decomposition and competition of *A. microsporum* and *S. pachcystidiatus* on cones of *P*. subgenus *Strobus*. The yellow pileus represents *S. pachcystidiatus*. (**f**) Successive decomposition and competition of *A. microsporum*, *S. pachcystidiatus* and *S. orientalis* on cones of *P*. subgenus *Strobus*.

**Table 1 jof-07-00679-t001:** The characteristics of the assembly scaffold and genomes of three *Auriscalpium* species and four *Strobilurus* species.

	*A. vulgare*	*A. microsporum*	*A. orientale*	*S. stephanocystis*	*S. pachycystidiatus*	*S. luchuensis*	*S. orientalis*
Strain No.	CBS 236.39	A-H-210-14	A-Y-A	CBS 113577	Y-H-6C	Y-Y-2D	K-1-1
Number of Contigs	104	54	38	38	71	58	43
Length of the genome assembly (Mb)	51.68	43.46	45.40	42.38	51.82	46.71	51.25
Contig N50 (Mb)	1.721	2.449	1.814	2.875	2.269	2.554	3.397
GC content (%)	56.39	56.13	56.68	52.13	51.75	51.76	51.53
Number of protein-coding genes	13,639	15,333	16,958	16,439	18,157	16,796	18,509
Median/Average gene length (bp)	1716/2202	1862/2246	1886/2300	1702/2033	1736/2139	1732/2116	1747/2146
Median/Average coding sequence size (bp)	1392/1675	1508/1801	1535/1867	1441/1721	1446/1754	1467/1761	1471/1787
Median/Average number of exons per gene	5/5.8	6/7.1	6/7.3	5/6.7	5/6.7	5/6.7	5/6.8
Median/Average exon size (bp)	172/288	144/252	140/257	143/256	148/263	150/265	146/262
Median/Average intron size (bp)	55/109	55/71	55/67	52/53	52/67	52/62	52/60
Median/Average size of intergenic regions (bp)	674/1544	399/1085	386/953	396/893	454/1086	462/1065	449/1046
Gene density (genes/Mbp)	264	353	374	388	350	360	361

**Table 2 jof-07-00679-t002:** Abundance and distribution of the top 10 groups of overexpressed genes in the presence of latex among the seven sequenced genomes in this study.

	InterPro Hit ID	InterPro Hit Description	*S. luchuensis*	*S. pachycystidiatus*	*S. stephanocystis*	*S. orientalis*	*A. microsporum*	*A. orientale*	*A. vulgare*	*Strobilurus*_mean	*Auriscalpium*_mean	Padj ^†^
1	IPR000209	Peptidase S8	6	4	5	4	19	23	20	4.75	20.67	0.021 *
2	IPR001461	Aspartic peptidase	17	17	16	19	18	34	28	17.25	26.67	0.432
3	IPR008972	Cupredoxin	8	8	7	7	9	5	7	7.5	7	0.889
4	IPR001128	Cytochrome P450	92	94	97	96	95	121	98	94.75	104.67	0.584
5	IPR018487	Hemopexin-like repeats	0	0	0	0	0	0	0	0	0	1
6	IPR015366	Peptidase S53	1	1	2	1	3	4	4	1.25	3.67	0.021 *
7	IPR001338	Hydrophobin	0	0	0	0	0	0	0	0	0	1
8	IPR001128	Cytochrome P450	31	22	27	37	20	23	28	29.25	23.67	0.432
9	IPR001338	Hydrophobin	4	9	4	11	6	6	6	7	6	0.876
10	IPR011701	Major facilitator superfamily	37	42	41	49	44	55	53	42.25	50.67	0.387

* significance at the level of *p* < 0.05. † adjusted *p*-value based on the False Discovery Rate (FDR) method for multiple testing correction.

**Table 3 jof-07-00679-t003:** Comparison of secondary metabolisms of fungi in *Auriscalpium* and *Strobilurus*.

Secondary metabolisms	*S. stephanocystis*	*S. luchuensis*	*S. pachycystidiatus*	*S. orientalis*	*A. vulgare*	*A. orientale*	*A. microsporum*	*Strobilurus*_mean	*Auriscalpium*_mean
Terpene	7	8	8	9	13	13	8	8	11.33
T1PKS ^†^-Terpene	0	0	1	0	0	0	0	0.25	0
T1PKS ^†^-NRPS-like	0	0	0	0	1	1	1	0	1
NRPS ^‡^-like	10	9	9	7	6	1	4	8.75	3.67
Siderophore	2	1	1	1	1	2	1	1.25	1.33
betalactone	0	0	0	0	1	1	1	0	1
Strobilurin	1	1	1	1	0	0	0	1	0
T1PKS ^†^	1	1	0	2	0	0	0	1	0
indole	1	1	1	1	0	0	0	1	0
total	22	21	21	21	22	18	15	21.25	18.33

† type I polyketide synthases. ‡ nonribosomal peptide synthetase.

## Data Availability

The genomic data of our study were deposited in DDBJ/EMBL/GenBank including the *A. vulgare* (Accession number: JAHBBC000000000; BioProject: PRJNA728955; BioSample:SAMN19107592), *A. microsporum* (Accession number: JAHLMG000000000; BioProject: PRJNA735801; BioSample: SAMN19598194), *A. orientale* (Accession number: JAHLMH000000000; BioProject: PRJNA735804; BioSample: SAMN19598576), *S. luchuensis* (Accession number: JAHLMJ000000000; BioProject: PRJNA735851; BioSample: SAMN19599093), *S. pachycystidiatus* (Accession number: JAHLMK000000000; BioProject: PRJNA735855; BioSample: SAMN19599128), *S. stephanocystis* (Accession number: JAHLMI000000000; BioProject: PRJNA735844; BioSample: SAMN19598582) and *S. orientalis* (Accession number: JAHLML000000000; BioProject: PRJNA735858; BioSample: SAMN19599131).

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
