# Peer review of "Genomic and Experimental Investigations of Auriscalpium and Strobilurus Fungi Reveal New Insights into Pinecone Decomposition"

_jof, 2021, doi:10.3390/jof7080679_

Round 1

Reviewer 1 Report

I believe that the authors are tackling a very novel topic and that their research provides very valuable information regarding association of saprophytes in litter decomposition in natural environments.

I think that the methodology is well presented and designed for the research question pursued.

The results are well presented, although there are some troubles reading some of the figures.

The discussion is appropriate to answer the research question proposed.

Overall, I think that the paper is very good and that it could be accepted as it is. If any, I would just review the figures to try to make them more clear for the readers. My review of the paper is very positive.

Author Response

I believe that the authors are tackling a very novel topic and that their research provides very valuable information regarding association of saprophytes in litter decomposition in natural environments.

Response:

Dear prof:

We appreciate your positive evaluation of our work and agree with the comments regarding the limitations of our study. Your summary of our work is precise. We are grateful to you for your effort reviewing our manuscript and your positive feedback. Thank you very much for your attention, evaluation and comments on our manuscript. We have revised the manuscript according to your kind advices and detailed suggestions.

I think that the methodology is well presented and designed for the research question pursued.

Response: Thank you very much.

The results are well presented, although there are some troubles reading some of the figures.

Response: Thank you very much. Some of the figures have been modified to facilitate readers' better understanding.

The discussion is appropriate to answer the research question proposed.

Response: Thank you very much.

Overall, I think that the paper is very good and that it could be accepted as it is. If any, I would just review the figures to try to make them more clear for the readers. My review of the paper is very positive.

Response: Thank you very much. Some of the figures have been modified to facilitate readers' better understanding.

Reviewer 2 Report

The authors present, to my knowledge, a novel study that aims to shed more light on mechanisms in fungal succession during decomposition in broader view and particularly in the case of pine cones in wider context. Coniferous forests are often underlain by soils that are poor in nutrients and decomposition of plant material including cones provides an important source of available nutrients that can by re-used by vegetation. From that reason, studies improving our knowledge of such decomposition are valuable.
In the introduction, authors provide a straight-forward, but comprehensive overview of current knowledge and I have no specific comments to this part.
In my opinion, the weakest part of the manuscript is the M&M section, as I find the description of the greenhouse planting experiment very insufficient. How many cones were used for each pine species/fungal pair? Where and when were the cones collected? Were there any criteria for the selection of collection site (stand composition, age, etc.) and cones (size etc.). Were the cones collected in comparable time? Was there any specific treatment of the cones between the collection and experiment start? When did the incubation start/end exactly? Are the conditions in the greenhouse stable throughout the whole experiment duration? Was there any mirroring of the changing temperature and/or humidity that would resemble natural changing of seasons? If no, could it somehow affect the succession? It is definitely driven by the changes in cone chemical composition, but could also the change in the climatic conditions throughout the year also play a role, as the occurrence of Strobilurus sp. rapidly increases in spring?
The other information I miss is how exactly was chosen the time when was assessed the chemical composition of "cones being decomposed by fungi in the genus Auriscalpium"? What was the time elapsed since the cone falling/collection/first Auriscalpium presence record? Was it exactly the same and comparable for all the pine species?
If I take into account that I miss some information in M&M that might have been reflected in results, all the other results present were clearly described.
Discussion was clear as well. However, I miss a bit deeper discussion in the section 4.2. Authors clearly demonstrate all possibilities of decomposition succession “routes” observed in the field. It would be very interesting if authors were able to provide an overview of how frequently do occur all of the possible ways (a-f) in Figure 5. How often it is that Strobilurus never appears on the cone and only Auriscalpium is present? Does always occurrence of Strobilurus mean disappearance of Auriscalpium? If not, how frequently and for how long can these fungi live simultaneously? Caption for Fig 5. states that it illustrates the types observed in the field. Did authors observe all the types in their greenhouse experiment? If it was not precisely recorded, is it possible to reciprocally estimate the presence and/or frequency of each type?
The manuscript presents outcomes of a very interesting study and is generally written in a sound style (as a non-native speaker, I will leave the judgment of English grammar correctness to experts). However, there are many uncertainties in the experiment design, that need to be clearly addressed before the acceptance.

Some specific comments:
Figure 1: The figure would be better self-explaining if the period of cone fall was illustrated (by a line or a bar) in the graphs of species occurrences.

Figure S1: Please better describe what is shown on y-axis. I didn’t get it.
Line 525: Correct “Fungual” to “Fungal”

Author Response

The authors present, to my knowledge, a novel study that aims to shed more light on mechanisms in fungal succession during decomposition in broader view and particularly in the case of pine cones in wider context. Coniferous forests are often underlain by soils that are poor in nutrients and decomposition of plant material including cones provides an important source of available nutrients that can by re-used by vegetation. From that reason, studies improving our knowledge of such decomposition are valuable.

Response:

Dear prof:

We appreciate your positive evaluation of our work and agree with the comments regarding the limitations of our study. Your summary of our work is precise. We are grateful to you for your effort reviewing our manuscript and your positive feedback. Thank you very much for your attention, evaluation and comments on our manuscript. We have revised the manuscript according to your kind advices and detailed suggestions. Here below we address the questions and suggestions raised by you.

In my opinion, the weakest part of the manuscript is the M&M section, as I find the description of the greenhouse planting experiment very insufficient.

Response: We are grateful for the suggestion. To be more clearly and in accordance with your concerns, we have added a more detailed interpretation regarding greenhouse planting experiment (lines 122-131).

How many cones were used for each pine species/fungal pair?

Response: In our study, we observed the succession of fungi in Auriscalpium and Strobilurus on three needle pine cones (such as Pinus yunnanensis and P. sylvestris) and five needle pine (such as P. armandii) under field conditions. Therefore, in order to show and record the succession process on cones, we selected the large cones of P. armandii with the number about 250 as samples to record and observe this process in greenhouse (lines 122-131).

Where and when were the cones collected?

Response: These cones were collected from Yufeng Temple, Lijiang City, Yunnan Province in July 2017 (lines 122-131).

Were there any criteria for the selection of collection site (stand composition, age, etc.) and cones (size etc.).

Response: The reason for the selection of collection site of Yufeng Temple is that Yufeng Temple is the distribution center of fungi in Strobilurus and Auriscalpium, and the climate in this area is close to Kunming City. Yufeng Temple has a mixed forest of P. armandii and P. yunnanensis. Five samples of fungi in Strobilurus and Auriscalpium sequenced by us were collected from Yufeng Temple. Moreover, we have collected and observed samples in this area for up to 10 years since the taxonomic study of Strobilurus and Auriscalpium. We first observed this phenomenon in the field, and then in order to further observe and record its succession, we brought the cones back to Kunming and cultured them in the greenhouse. We selected two states of cones, one is the newly fallen cones without fungus growth, and the other is the cones with fungi in Auriscalpium growth (lines 122-131, page 3).

Were the cones collected in comparable time?

Response: These cones were collected in Yufeng Temple in Lijiang City in the same period in July 2017 (lines 122-131, page 3).

Was there any specific treatment of the cones between the collection and experiment start? When did the incubation start/end exactly?

Response: We brought back cones in two states from Yufeng Temple. One is the cones without any fungus growth, and the other is the cones with only A. microsporum growth. The cones were not specially treated. After we brought it back, we put it in the plastic shed and spray water regularly to ensure its humidity, so as to observe the growth of fungi (lines 122-131, page 3).

Are the conditions in the greenhouse stable throughout the whole experiment duration? Was there any mirroring of the changing temperature and/or humidity that would resemble natural changing of seasons?

Response: During our experiment, we keep the temperature in the greenhouse consistent with the outdoor temperature in Kunming City. In order to ensure its humidity, we spray water regularly. We found that its succession law basically has little change with the field conditions, but because the environmental humidity in the greenhouse is stable, its fruiting time is relatively continuous. In the natural state, the fruiting time is greatly affected by the environmental humidity, more mushrooms are produced when it is wet, no or less mushrooms are produced when it is dry (lines 122-131, page 3).

If no, could it somehow affect the succession? It is definitely driven by the changes in cone chemical composition, but could also the change in the climatic conditions throughout the year also play a role, as the occurrence of Strobilurus sp. rapidly increases in spring ?

Response: In the field observation, we found that the succession on the cone is mainly the order of those two fungi. We obtained the same results under the field observation and laboratory conditions. Our observation results and greenhouse culture results show that although humidity and temperature will have a certain impact on the fruiting time, neither of them can determine the preference of fungi in Auriscalpium and Strobilurus for substrates with different degrees of decay, that is, fungi in Auriscalpium appear on newly fallen cones, and fungi in Strobilurus appear on cones with high degree of decay.

The other information I miss is how exactly was chosen the time when was assessed the chemical composition of “cones being decomposed by fungi in the genus Auriscalpium”? What was the time elapsed since the cone falling/collection/first Auriscalpium presence record? Was it exactly the same and comparable for all the pine species?

Response: We choose two states for the measurement of cone composition, one is the newly fallen cone, and the other is the cone decomposed by the Auriscalpium and just colonized by Strobilurus (Figure S3) (Lines 144-146, page 4).

In our observation, the fungi in Auriscalpium can decompose independently without fungi in Strobilurus. For the cones of three needle pine (P. yunanensis and P. sylvestris, etc.), this succession processes usually last about 2-3 years, and for the cones of five needle pine (P. armandii), it usually lasts about 3 years. If there are fungi in Strobilurus, due to the competition of Strobilurus, the fungi in Auriscalpium are usually excluded by the competition of fungi in Strobilurus, so the decomposition time of Auriscalpium is usually shortened. Because the size of cones and the number of Auriscalpium growing on cones, the temperature and humidity are different, so the decomposition time may not be exactly the same under field conditions, but their occurrence order will not be reversed, that is, the fungi in Auriscalpium always appears on the fresh cone, and the fungi in Strobilurus always appears on the cone decomposed by fungi in Auriscalpium.

If I take into account that I miss some information in M&M that might have been reflected in results, all the other results present were clearly described. Discussion was clear as well. However, I miss a bit deeper discussion in the section 4.2. Authors clearly demonstrate all possibilities of decomposition succession “routes” observed in the field. It would be very interesting if authors were able to provide an overview of how frequently do occur all of the possible ways (a-f) in Figure 5.

Response: In fact, it is difficult for us to make statistics. For example, in some regions (such as the tropics), we only observed the fungi in Auriscalpium on cones. Because fungi in Strobilurus are hardly distributed in the tropics. In some regions where the two genera grow at the same time, the fungi in Auriscalpium first appear on the cones, and after the cones are decomposed for a period of time by fungi in Auriscalpium, fungi in Strobilurus grow on it.

How often it is that Strobilurus never appears on the cone and only Auriscalpium is present ?

Response: This can happen in the tropics. However, in the area where fungi in Auriscalpium and Strobilurus grow together, or in our cultivation experiments, almost 100 % fungi in Auriscalpium will be competed by fungi in Strobilurus in the late stage of the decomposition of cones.

Does always occurrence of Strobilurus mean disappearance of Auriscalpium? If not, how frequently and for how long can these fungi live simultaneously? Caption for Fig 5. states that it illustrates the types observed in the field. Did authors observe all the types in their greenhouse experiment? If it was not precisely recorded, is it possible to reciprocally estimate the presence and/or frequency of each type?

Response: In fact, during the competition, the Auriscalpium and Strobilurus often appear on the same cone (Figure S5), and sometimes we may not observe this phenomenon, because sometimes the fungi in Auriscalpium may not produce fruit bodies at the end of the decomposition of the cones. We find that the length of this process is different, because the number of fruit bodies of fungi in Auriscalpium or Strobilurus growing on the cone is not certain, which may be caused by a variety of factors. The period of their co-existence on the cone is not constant, which may also be related to the size of the cones, the number of fruit bodies of fungi in Auriscalpium and Strobilurus, temperature and humidity, fruiting time, etc.

In our greenhouse, we observed the succession of these three fungi on cones, because these three cones are distributed in the same domain. These cones were collected from Yufeng Temple in Lijiang City, that is, before we collected them, the spores of these three species may already existed on the cones. However, these three species are not distributed in all collection areas. For example, in Yeya Lake in Kunming City, only two fungi, A. microsporum and S. orientalis, can be collected, and the succession of these two fungi only occurs on cones. In some areas in Northwest Yunnan, only two fungi, A. microsporum and S. pachcystidiatus, can be collected, Accordingly, only the succession of these two fungi occurred on the cones.

The manuscript presents outcomes of a very interesting study and is generally written in a sound style (as a non-native speaker, I will leave the judgment of English grammar correctness to experts). However, there are many uncertainties in the experiment design, that need to be clearly addressed before the acceptance.

Response: We are grateful to you for your effort reviewing our paper and your positive feedback. Your summary of our work is precise. We deeply appreciate your suggestion. According to your comment, we have provided more details to describe the possible reasons.

Some specific comments:

Figure 1: The figure would be better self-explaining if the period of cone fall was illustrated (by a line or a bar) in the graphs of species occurrences.

Response: Cones of P. yunnanensis, P. massoniana, P. sylvestris and P. armandii will fall continuously throughout the year, rather than directly when they are mature. It seems that the fruiting time of fungi in Auriscalpium is not affected by the falling time. In the mushroom season of fungi in Strobilurus, there are still a large number of newly fallen cones on the ground, while in the fruiting time of fungi in Auriscalpium, there are still highly rotten cones, but there is no Auriscalpium.

Figure S1: Please better describe what is shown on y-axis. I didn’t get it.

Response: For the graphs, the x-axis shows different species and the y-axis shows number of occurrences of collected or observed fruiting bodies on wood in different years in Fenglin Nature Reserve. Different colors represent different levels of decay, and lengths represent numbers.

Line 525: Correct “Fungual” to “Fungal”

Response: Corrected. Thank you very much.

Reviewer 3 Report

The manuscript Genomic and Experimental Investigations of Auriscalpium and Strobilurus Fungi Reveal New Insights into Pinecone Decomposition ” by Panmeng Wang, Jianping Xu, Gang Wu, Tiezhi Liu and Zhuliang Yang describes the sequencing of the whole genome of seven fungi isolated from cones of pines. Genome sequencing was performed along with greenhouse experiments. Several other representatives including members of the same families have been sequenced in the last years providing substantial information for comparative studies. On the other hand, the succession during pinecone decomposition has not been studied so far. All analyses and searches are technically sound and the text is clearly written. However, it requires minor changes. Methods should have versions and parameters given for reproducibility.

genomes

Obtained genome completeness was assessed with BUSCO and the assembly achieved is almost complete with 38-104 contigs. Annotation was trained on related taxa. 

I did not find a transposon masking step in the annotation pipeline. Unmasked transposons can be recognized as genes, which could lead to the elevated gene numbers observed in Table.1. It is highly recommendable to re-run the gene calling on a masked genome. More up to date information on mobile elements and a community working on transposons can be found here https://tehub.org/.

There are BioProject numbers provided but I could not find assembly id and annotation ids in the data availability section. In order to comply with data management policies, these should be added to the final version of the manuscript. 

Functional annotation

Functional annotation can be improved using tools to map on KEGG - it is recommendable to run Kofamscan. CaZy cover only some of the enzymes involved in carbohydrate metabolism, and the obtained resources (complete genomes) enable the authors to have a more complex view of the whole metabolism of Auricalpium and Strobilurus fungi. Moreover, expansions and contractions of gene families can be studied in a framework like Cafe to show how the protein family size changed on the evolutionary tree of the analysed taxa. Noteworthy a low copy number of a certain family does not necessarily predict low enzymatic activity.

The elevated numbers of S8 and S53 peptidases can have broader significance than resin usage as a carbon source. Maybe they are used to feed on invertebrates to complement the diet?

Antismash has now v.6 and it is worth checking if predictions of SMC are the same with the upgraded software. I was wondering whether the betalactone clusters in Auriscalpium are of bacterial origin. What type of indole clusters are there, is there a compound associated with these clusters?

Authors placed their findings in the context of other taxa and phenotypic observations were coupled with gene content. The fungi analysed display great differences in physiology what sets the ground for further experimental studies.

The text would benefit from listing unanswered questions eg. the role of associated bacteria and possible future research directions.

Author Response

Several other representatives including members of the same families have been sequenced in the last years providing substantial information for comparative studies. On the other hand, the succession during pinecone decomposition has not been studied so far. All analyses and searches are technically sound and the text is clearly written. However, it requires minor changes. Methods should have versions and parameters given for reproducibility.

Response:

Dear prof.:

We are grateful to you for your effort reviewing our manuscript and your positive feedback. The summary of our work as written by you is precise. Here below we address the questions and suggestions raised by your comments. We have carefully addressed all your concerns. Please see below our replies. We hope you are satisfied with our answers and the new figure or data we provided.

Genomes:

I did not find a transposon masking step in the annotation pipeline. Unmasked transposons can be recognized as genes, which could lead to the elevated gene numbers observed in Table.1. It is highly recommendable to re-run the gene calling on a masked genome. More up to date information on mobile elements and a community working on transposons can be found here https://tehub.org/.

Response: There is a transposon masking step in our analysis using the software of GETA (https://github.com/chenlianfu/geta), which combines RNA-aided annotation, homology searches, and de novo prediction. Firstly, repeat-masked genome assembly was obtained by using RepeatMasker based on repeat sequences identified with RepeatModeler. Secondly, the next-generation clean reads were aligned to the genome sequences using HISTA2, and then genes were predicted based on the open reading frame (ORF) of the optimal transcripts. Thirdly, homologous annotation was conducted by searching genome contigs against protein sequences of related species with BLAST, followed by Genewise annotation. Fourthly, de novo annotation was performed using AUGUSTUS. Finally, the above three gene annotations were integrated to obtain the final result [42]. Because analysis method of GETA software has been introduced in detail in his article, we only cite his article in our manuscript without introducing the method in detail.

There are BioProject numbers provided but I could not find assembly id and annotation ids in the data availability section. In order to comply with data management policies, these should be added to the final version of the manuscript.

Response: We are extremely grateful to reviewer for pointing out this problem. When the manuscript is accepted, we will release the genome sequence. In addition, we submit the gff3 files on the attachments.

Functional annotation: Functional annotation can be improved using tools to map on KEGG - it is recommendable to run Kofamscan. CaZy cover only some of the enzymes involved in carbohydrate metabolism, and the obtained resources (complete genomes) enable the authors to have a more complex view of the whole metabolism of Auricalpium and Strobilurus fungi.

Response: Thank you for your precious comments and advice. We have made functional annotation including Nr, Go, KEGG, KOG, and swiss-prot. Due to the large amount of data, it is not fully displayed. Some results will be published in subsequent articles. This time, we will take the results of eggo's genome annotation (including KEGG annotation) on the attachments for readers' reference.

Moreover, expansions and contractions of gene families can be studied in a framework like Cafe to show how the protein family size changed on the evolutionary tree of the analysed taxa. Noteworthy a low copy number of a certain family does not necessarily predict low enzymatic activity.

Response: We are extremely grateful to reviewer for pointing out this problem. According to your comment, we have provided the analyses of expansions and contractions of gene families using Cafe (line 220-238; line 454-478; line 538-548; line 565-579).

The elevated numbers of S8 and S53 peptidases can have broader significance than resin usage as a carbon source. Maybe they are used to feed on invertebrates to complement the diet ?

Response:We are grateful for the suggestion, however, I did not fully understand the meaning of this sentence. Peptidases S8 and S53 are among the top 10 most up-regulated enzymes in R. microporus in the presence of latex [50, Table 2]. S8 and S53 may be related to the decomposition of proteins in the latex, causing it to peel off [50]. The most compelling evidence supporting the early colonizing ability of Auriscalpium fungi is that their number of peptidases S8 and S53 is far greater than that in Strobilurus fungi. The genomic evidence is consistent with Auriscalpium fungi capable of colonizing newly fallen cones and decomposing proteins in resin rapidly.

Antismash has now v.6 and it is worth checking if predictions of SMC are the same with the upgraded software. I was wondering whether the betalactone clusters in Auriscalpium are of bacterial origin. What type of indole clusters are there, is there a compound associated with these clusters?

Response: Because we annotate in the web version, which uses the latest database. In order to make readers understand the annotation information more clearly, we will upload the specific information of the annotation as an attachment. In addition, indole alkaloids are closer to ergot alkaloids according to the annotation results.

Authors placed their findings in the context of other taxa and phenotypic observations were coupled with gene content. The fungi analysed display great differences in physiology what sets the ground for further experimental studies. The text would benefit from listing unanswered questions eg. the role of associated bacteria and possible future research directions.

Response: I am very grateful to your comments for the manuscript. For future research directions, we will clearly delineate successional timelines between Auriscalpium and Strobilurus to inform the mechanisms of successional decomposition through controlling expression profiles based on these timelines. Further associating these profiles with realized relative loss in the cones over the time course has the potential to link critical successional dynamics to functional outcomes (line 656-660).

Round 2

Reviewer 2 Report

The authors provided a very sound response to my comments, explained all uncertainities and revised the manuscript accordingly. I do not see any obstacle for acceptance.